# Selection of TCC Curve and Protection Cooperation Method of Distribution Line Using Linear Optimization

**Geonho Kim** [1] , **Woo-Hyun Kim** [1] **and Chun-Kwon Lee** [2],*

1   Distribution Laboratory, Power Korea Electric Power Corporation Research Institute,
    Daejeon 34056, Republic of Korea; ghkim@kepco.co.kr (G.K.); wh.kim@kepco.co.kr (W.-H.K.)
2   Control and Instrumentation Engineering, Pukyong National University, Busan 48547, Republic of Korea
*   Correspondence: ck.lee@pknu.ac.kr

**Abstract:** Distribution systems are mostly composed of radial structures, which are susceptible to an increased variability and complexity of system operation due to frequent line changes during operation. When multiple changes in distribution lines occur simultaneously, the relative positions of protective devices also change. The existing protection coordination method of distribution lines is configured by considering the operation characteristics and coordination time interval (CTI) of all protective devices in series from the substation to the terminal load. Therefore, the protection coordination algorithm needs to be redesigned whenever a line is changed or a protective device is added to the distribution line for which the existing protection coordination algorithm has been set. In addition, existing protection coordination methods require complex calculations and procedures, which are subject to human errors and are less feasible for responding in real-time to changes in the distribution system. In this paper, we propose the adaptive time–current curve (TCC) method by selecting the time dial setting (TDS) and minimum response time (MRT) of individual protective devices in accordance with the relative distance based on the linear optimization technique. Using PSCAD/EMTDC, a power system analysis program, the minimum operating current and the fault current of each protective device are obtained, and the proposed protection coordination algorithm is verified according to the series configuration relationship of the protective devices. Finally, the proposed method is applied to an actual distribution line to verify the improvement over the existing protection coordination.

**Keywords:** protection coordination; parameter optimization; time–current characteristics

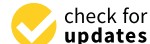



## 1. Introduction

In recent years, the need for reliable operation and protection technologies for various power distribution systems has increased to ensure a stable and high-quality power supply to consumers. Figure 1 shows the statistics of the system average interruption duration index (SAIDI) compared to investments related to the distribution line protection coordination from 2010 to 2018 in South Korea [1]. Since 2010, capital investment in protection devices has steadily increased, roughly tripling. As a result, the average peak load between protection devices and SAIDI has steadily decreased. However, relative to the cost of investment, the reduction in damage caused by power outages is becoming saturated and less effective. Rather than installing and operating a large number of power protective devices to minimize outage damage, domestic and foreign electric power companies need to improve distribution protection technology to promote stable grid operation and economic efficiency.

A fault occurrence during power grid operation poses safety hazards of property destruction and human injuries due to the exposure of heat generated by arcing and may further inflict damage to power equipment. Therefore, a rapid but precise determination of the fault location is necessary to minimize the fault section [2]. A single distribution line originating from a substation is subjected to a complex mixture of various geographical and

environmental characteristics on its way to its terminals. Therefore, each of the versatile characteristics should be considered to establish an optimal protection coordination system to minimize the number of fault sections occurring on the distribution line [3]. If there is more than one protective device in a distribution line, only the device in closest proximity to the fault location should be opened, while the other protective devices remain intact [4,5].

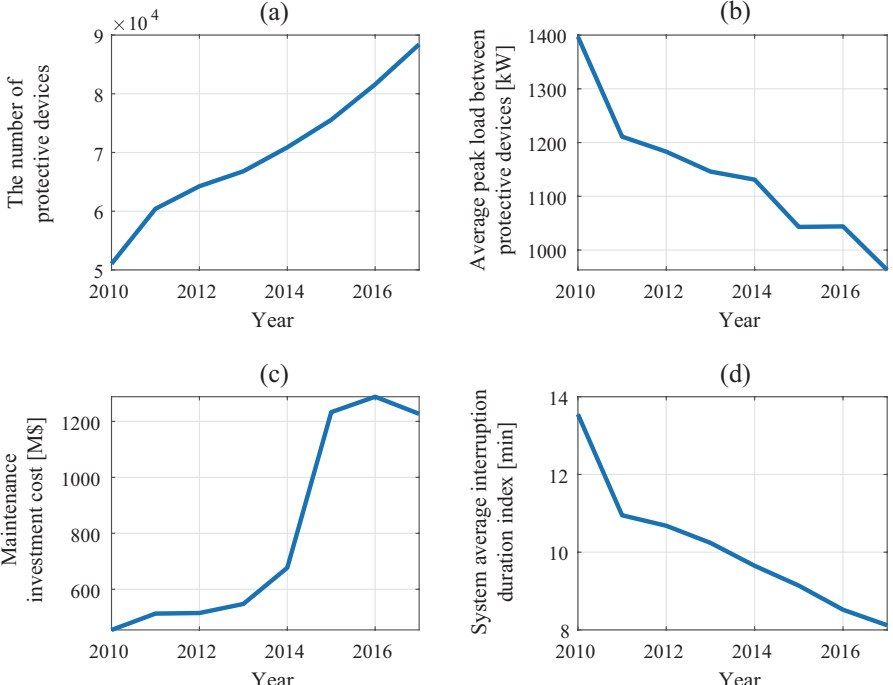

**Figure 1.** Distribution line maintenance statistics in South Korea: (**a**) the number of protective devices, (**b**) average peak load between protective devices, (**c**) maintenance investment cost and (**d**) system average interruption duration index.

In a distribution system, the recloser (R/C) is used for overhead lines, and the multi-circuit breaker automated (MCA) is used for protection coordination in underground distribution lines. The epoxy fault interrupter (EFI) circuit breaker is used to block fault currents from entering areas of high customer density or distribution generators (DGs). R/C sequentially performs two time–current curves (TCCs) for phase or ground faults. It divides into an instantaneous operation with a fast-operating time (F) followed by a long delay operation (D) and performs the blocking and reclosing of the fault section with 2F2D sequence operations. The most important goal of protection cooperation is to precisely block solely the faulty section by matching the cooperation interval time (CTI) so that protection devices do not operate simultaneously.

Globally, a large number of small DGs are being connected to the distribution system. In addition, the design, operation and management of the distribution system are growing in complexity, eventually posing various problems for protection coordination [6]. In particular, as many DGs are connected when a fault occurs in separate distribution lines connected to the same main transformer (MTr), the protective device malfunctions due to the reverse fault current from the DG, or the protective device may not operate regardless of whether a fault occurs in the line connected to the DG due to the apparent effect. To prevent this, directional relays and incoming circuit breakers (CBs) are required [7].

In addition, as the volatility of supplied power increases according to the output of DGs, the power grid operation is made more complex by frequent changes in line configuration operation. The conditions under which distribution lines are switched can be divided into four main categories. First is when the power supply of a particular section is shared by other lines that can spare it, depending on the line load and the generation of DGs. Second, temporary changes are made to minimize power outages due to repair

work on distribution lines. Third, in the event of a distribution line failure, the normal section is connected to other lines to minimize the faulty section. Lastly, there is a change in the connection position of existing lines due to the construction of new distribution lines. Excluding the last condition, changes in distribution lines are made from time to time and may occur simultaneously. These alterations in distribution lines may further entail the relocation of the protective devices installed within the section. When a change occurs in the configuration of the protection coordination algorithm or additional protective devices are connected in series, the operation settings and CTI of all devices on the changed lines should be readdressed [8]. Protection coordination operators in the distribution system are faced with the practical difficulty of performing frequent protection coordination assessments whenever a line is changed; thus, a technical solution is needed.

A.F. Naiem et al. presented an appropriate protection coordination model that reduces the number of impracticable protection cases for a distribution line with DG [9]. The method yields ideal results through assumptions of the DG and R/C locations and multiple protection coordination simulations at each fault location. However, limitations exist when applying the model to contemporary distribution lines subjected to multiple line changes induced by dispersed generation and volatile load supply and demand. S. Chaitusaney et al. reanalyzed cases of protection coordination failures in distribution systems with DG penetration [10]. The group focused on the effect of changes in DG capacity on the fault current, for cases in which DG is connected to a complex distribution system. Study results suggest that a feeder breaker is necessary to minimize the impact of fault occurrence within distribution systems with DG penetration exceeding a certain capacity. Moreover, the existing protection coordination scheme should be rendered to enable protection coordination during the feeder breaker period. A.Y. Abdelaziz et al. proposed an adaptive protection coordination method that enables the selection of an optimal time dial setting (TDS) through the linear optimization of TCCs of distribution system protection devices [11]. Following TDS linear optimization, a suitable protection coordination algorithm is proposed when the supervisory control and data acquisition (SCADA) system recognizes a change in the shape and configuration of the distribution line leading to a consequent change in breaker position. The study greatly contributed to the analysis and linearization of complex characteristic curves for protection coordination. Conversely, incorporating a minimum operation time that takes into account the CTI of the protective device resulted in limitations regarding the execution of reduced linear optimization. Therefore, the customer relay or fuse may not operate properly if a minimized TDS value is applied to the distribution line's breaker. As a result, locating a fault may be challenging, or the fuse may fail to operate, leading to increased coverage of the faulty area. Studies by H. Muda and M. Shih analyzed characteristic curves to derive appropriate values for the setting of overcurrent relays [12,13]. A major drawback of the studies is that a setting method for the cooperation of multiple protective devices installed in the distribution line is not included, since the behavior characteristics of cooperation and the reclosing of protective devices are not considered. Therefore, this method is only applicable to the consistent setting of protective devices.

In this paper, we propose an intuitive protection coordination method for power distribution systems through the analysis of the series configurations and operation characteristics of protection devices. Section 2 analyzes the TCC characteristic of the IEC standard and the operation characteristics of protective devices and examines detailed factors such as the CTI, protection cooperation range according to fault current and appropriate pick-up current. In Section 3, the criteria and constraints in Section 2 are used to select the appropriate TDS for individual protective devices by applying the linear minimum mean square error (LMMSE) estimation. Then, the minimum response time (MRT) is applied to ensure an accurate protection CTI regardless of the magnitude of the fault current. The proposed protection coordination algorithm is applied to the distribution overhead line model to verify the performance of the protection coordination for a single-line ground fault (asymmetrical fault) and a three-phase short circuit fault (symmetrical fault). In Section 4,

an empirical test is conducted to verify the proper protection coordination by simulating a ground fault utilizing an artificial fault generator (AFG) in an actual power distribution system. Finally, Section 5 concludes this works.

## 2. TCC Selection and Analysis for Constraints on Protective Devices

### 2.1. Analysis of Time Difference between Protection Cooperation between Devices

When designing the protection coordination algorithm, the CTI between protective devices of the power side and load side must be considered. This is because the blocking operation time derived from the characteristic curve does not reflect the physical characteristics of the actual protective device. In order for the protective device to operate, the relay first detects the fault current and subsequently sends a signal to the CB. Thereafter, a time delay occurs for the CB to operate and extinguish the arc.

The CTI includes fault current detection, signal transmission, CB operation and arc extinguishing time. Since CTI may vary depending on the product or manufacturer, it is necessary to know in advance the CTI characteristics of the distribution line protective device to perform accurate protection coordination. In the case of an analog relay, the induction disc rotates in response to the fault current and transmits a signal that determines the cut-off time. Moreover, the return time should be reflected after the fault has been cleared. On the other hand, digital relays do not consider the return time due to the characteristics of electronic equipment [14].

Until the majority of protective devices in distribution lines have switched to digital relays, some substations will still use analog relays. Therefore, a maximum of 10 cycles was applied for the CTI between substation CB and distribution line protective devices. If all relays are changed to digital in the future, the CTI between the substation breaker and R/C can be applied as 3 cycles [14]. Considering that R/C also has a model that uses an analog relay, the CTI was set to 3.5 cycles considering the difference in cooperation time. The CTIs between different protective devices are described in Table 1.

**Table 1.** CTI of substation CB and protective devices.

| Power Side Protective Device | Load Side Protective Device | CTI (Cycle) |
|---|---|---|
| Substation CB | Analog/Digital relay R/C | 10 |
| Analog relay R/C | Analog relay R/C | 3.5 |
| Analog relay R/C | Digital relay R/C | 3.5 |
| Digital relay R/C | Analog relay R/C | 3 |
| Digital relay R/C | Digital relay R/C | 3 |
| Analog relay R/C | EFI | 3.5 |
| Digital relay R/C | EFI | 3 |
| MCA | MCA | 3 |

When designing the protection coordination algorithm, the CTI of protective devices should exceed that of Table 1. If a cooperative time difference less than CTI is applied to the protective device, both the power side and the load side protective devices are cut off by the fault current, which causes difficulty in determining the location of the fault and results in an extended power outage range.

### 2.2. Distribution Line Measurement Error and Fault Current Range Analysis

In order to increase the reliability of the protection coordination algorithm, the measurement error of the current transformer (CT) should be incorporated into the protection coordination algorithm design. The expression "5P20" is the common CT class according to the IEC standard. The number "20" denotes the accuracy limit factor (ALF), which indicates the maximum current level allowed to flow through the CT's core without reaching

over-saturation. For instance, if the CT ratio is 1000/5, the primary current via the CT can reach a maximum of $20 \times 1000 = 20$ kAs. The CT's standard ALF is 10, 15, 20 and 30. The measurement accuracy of CT is indicated by the number "5". The CT reads within the composite error of 5% when the current flowing through the "5P20" protection class CT is 20 times the rated main current [15]. A small ALF results in the CT secondary side current recognizing a current smaller than the actual fault current, and thus the protective device operates very slowly. When the IEC R/C standard is 5P20, the following conditions must be satisfied [16]:

(A)　$\pm 1\%$ measurement error within the rated value (600 A),
(B)　$\pm 5\%$ measurement error up to 20 times the rated value (12,000 A),
(C)　Apply $\pm 10\%$ by adding a design margin of $\pm 5\%$ in the (B) to ensure stable and safe operation in the high current area.

According to distribution operation standards, protective devices are generally installed at a distance exceeding 2 km from the substation [14]. Therefore, 2 km from the substation was selected as the location for the first protective device installation. Figure 2 depicts the standard fault current according to the distance to the overhead line and the underground line based on a distribution line with general capacity. As impedance increases with the length of the line, the fault current decreases. In Figure 2, the maximum fault current is about 6500 A of short-circuit current and about 5800 A of ground fault current in the underground distribution line. Therefore, protection cooperation between devices against maximum ground and short-circuit faults is required. These fault currents were selected as they can occur in the first protective device. Based on the corresponding fault current, the operating range of the distribution line protective device can be calculated to derive a protection coordination setting value within an appropriate range.

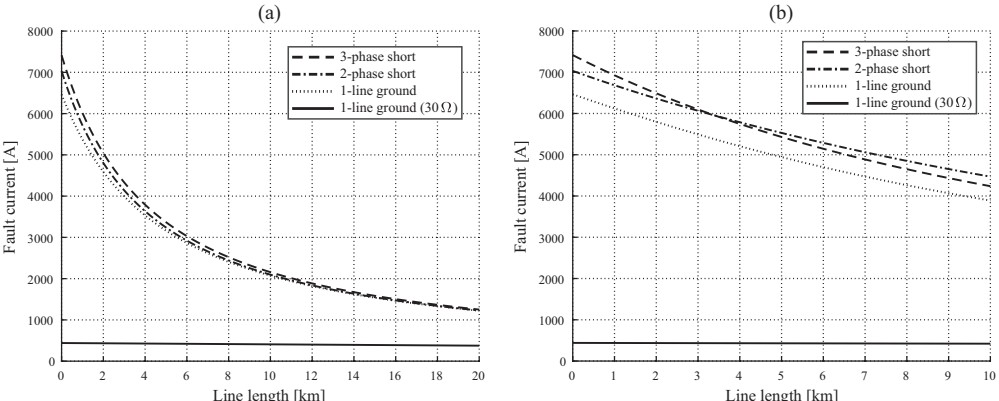

**Figure 2.** Fault current according to the line length: (**a**) overhead line (ACSR 160 mm$^2$) and (**b**) underground line (CNCV 325 mm$^2$).

*2.3. TDS Range Selection for Substation Relay*

The TDS range of the TCC is determined by the minimum operating current and the fault current according to the configuration of the distribution line, satisfying the minimum time of blocking operation for each fault. For example, relays of distribution lines in South Korea apply a proprietary KEPCO very inverse TCC curve based on ANSI as follows [17,18]:

$$t = \left( \frac{3.985}{\left( \frac{I_f}{I_p} \right)^{1.95} - 1} + 0.1084 \right) \times TDS \qquad (1)$$

The operating capacity of distribution lines is divided into regular and emergency operating capacities. In this study, an appropriate setting range of substation relays is calculated based on the emergency operating capacity (14,000 kVA/352 A at 22.0 kV).

Since the bus equivalent impedance of substations and the grounding system of MTr are standardized for operation, most substations have a similar fault current. Therefore, it is assumed that the magnitudes of the fault currents generated in the busbars on the power side of the distribution system are almost the same. Based on this assumption, a constant set value can be given to the substation relay configuration procedure according to the maximum load current and the fault current. Through the above process, after calculating the universal fault capacity of the distribution system, an appropriate time-delayed TDS value of the substation relay is derived. Subsequently, the TDS of the substation relay can be designed by calculating the maximum fault current of the power supply side. Then, the TDS range of the distribution line protective devices and terminal relay for customer load can be specified. The three-phase short circuit fault current and the single-line ground fault current of 100 domestic substations are illustrated in Figure 3.

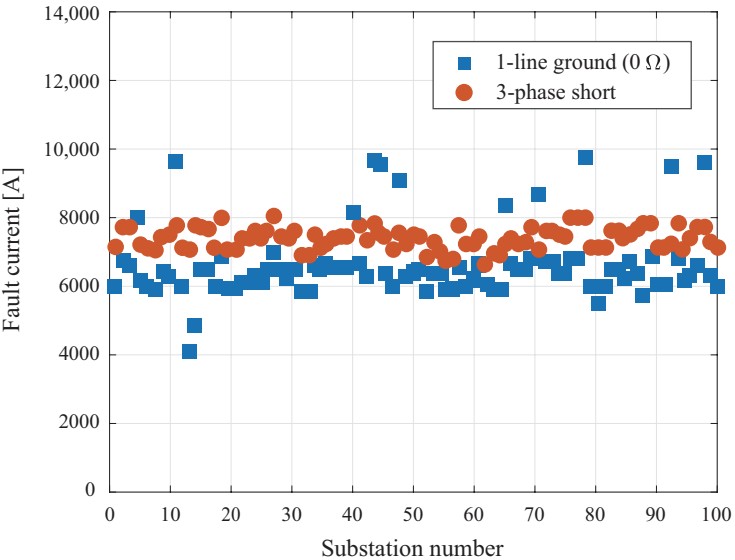

**Figure 3.** Short-circuit and ground fault current of the substation draw-out point.

The three-phase fault currents were 7000∼8000 A, which is an average current of about 7500 A, and the single-line ground fault currents were 6000∼7000 A with an average of 6500 A. The TDS of the substation relay for the three-phase short-circuit and single-line ground faults are calculated in Table 2, respectively [14,17–19]. The CT ratio of the substation relay was based on 600/5 CT.

Therefore, the time-delayed taps of OCR ($I_p$) of OCR and OCGR were set to 4.5 and 1, respectively. With the blocking operation time based on the pick-up current applied to each tap, the TDS calculation for the conditions in Table 2 is as follows:

$$t_{OCR} = \left( \frac{3.985}{\left( \frac{I_f}{4.5 \times 120} \right)^{1.95} - 1} + 0.1084 \right) \times TDS \leq 0.5 \text{ s}$$

$$t_{OCGR} = \left( \frac{3.985}{\left( \frac{I_f}{1.0 \times 120} \right)^{1.95} - 1} + 0.1084 \right) \times TDS \leq 0.5 \text{ s}$$

(2)

According to the maximum–minimum fault current ranges in Figure 3, the TDS range of the three-phase short circuit current is 3.57 (fault current: 6500 A) to 3.93 (fault current: 8500 A), and the TDS range of a single-line ground fault current is 4.51 ( Fault current: 5500 A) to 4.57 (fault current: 8000 A). Since the variation of TDS according to the fault current is small, the TDS value of the substation relay can be determined in the emergency operation distribution line by applying the TDS value calculated based on the average fault

current. Consequently, the TDS for three-phase short-circuit faults was selected as 3.78, and the TDS for single-line ground faults was chosen as 4.5. Since the TDS were set based on the emergency operating capacity, the tap and TDS can be selected above the set values in the regular operating capacity. After obtaining the substation TCC located at the top of the protection coordination TCC, the coordination time of the distribution line protective devices (R/C, EFI, MCA, etc.) located on the load side, in turn, can be determined.

**Table 2.** Relay configuration guide of a substation.

| | | Relay Configuration | |
|---|---|---|---|
| Time delayed Tap | OCR | Maximum operating current (352 A) × 1.5/CT ratio (600/5) = 4.41 A < Tap | |
| | OCGR | Maximum operating current (352 A) × 0.3/CT ratio (600/5) = 0.88 A < Tap | |
| Time delayed TDS | OCR | Blocking operation less than 0.5 s in case of three-phase short fault | |
| | OCGR | Blocking operation less than 0.5 s in case of single-line ground fault | |

### 2.4. Selection of Adequate Pick-Up Current of Protective Device

In an overhead distribution line, one to three R/Cs are usually operated in series. If EFI is added for a quick cut-off of the end load section, four protective devices are operated in series (three R/Cs and one EFI). Therefore, the difference between the pick-up current of the protective device must be set to ensure accurate protection coordination. A small pick-up current results in a reduced operation time of a protection device. Therefore, as shown in Figure 4, if located further from the power side, the pick-up current must be set to a smaller value to cut off first in case of a failure [14].

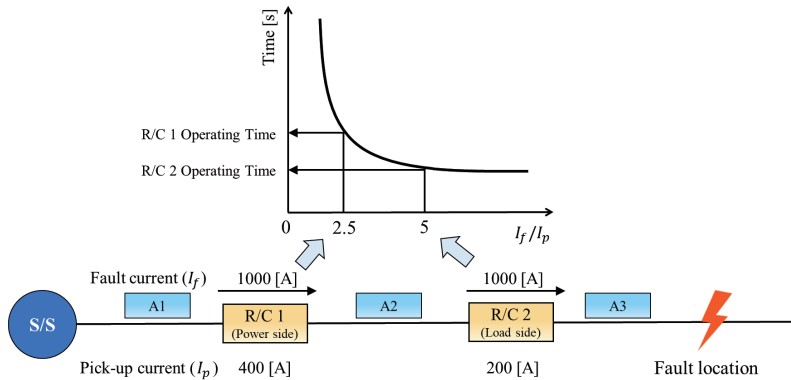

**Figure 4.** Protective device operation according to pick-up current.

The protective device setting and configuration guideline do not include a separate regulation for setting the pick-up current difference for the protective devices in series. Therefore, the protection coordination operators empirically select the setting value due to an absence of a consistent standard. When operating the 10 MVA standard distribution line with uniform load distribution, the load current flowing through the series-connected protective devices ($I_{nk}$) can be expressed as follows:

$$I_{nk} = \frac{n+1-k}{n+1} \times I_{max} \tag{3}$$

where $n$ is the total number of the connected distribution protective devices, $k$ is the serial order of the protective devices, and $I_{max}$ is the maximum load current of distribution lines (252 A at 10 MVA). Pick-up current selection in a series configuration of three R/Cs in a distribution line with equal loads is shown in Figure 5. Depending on the location, the load current of each protective device decreases, and the phase and ground pick-up currents are calculated.

Accurate protection coordination is possible within the 25% difference between CB and R/C and the 33% difference between R/Cs in the pick-up current with an equal load

distribution line. The pick-up current is calculated only with the load current. Therefore, the pick-up current may differ according to load distribution, topology and impedance.

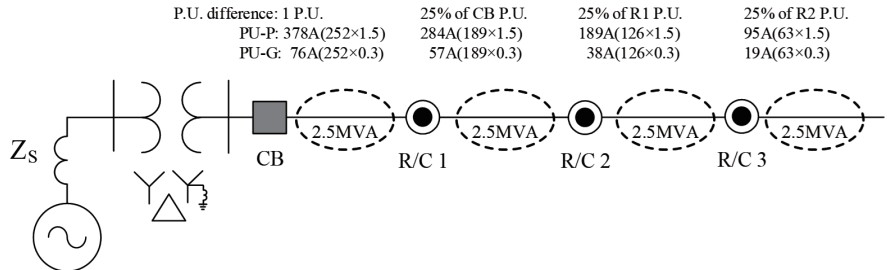

**Figure 5.** Example of correcting protective device for four-division equal load distribution line.

*2.5. Minimum Unit Selection of TDS and MRT*

A commonly implemented TDS setting range of protective devices for distribution systems is 0.10 to 2.00, and the minimum unit for change is 0.10 Step. An EFI is installed at the end of the distribution line to block the incoming fault current from flowing into the DG. Therefore, the TDS of EFI was set to 0.05—the lowest unit. In addition, the minimum response time (MRT) was applied to secure the CTI in the instantaneous fault current area. The protective device must operate faster than the instantaneous operation of the substation CB, and in order to maintain the CTI between the protective devices in Table 1, 60 ms (3.6 cycles) was collectively applied, and 20 ms was added to the protective device at the end. Then, 50 ms MRT was applied to coordinate the instantaneous fault current area of customer relays and power side protective devices. Table 3 summarizes the MRT setting results for each series of device configurations.

**Table 3.** CTI of substation CB and protective devices.

| Type | MRT Setting | | | |
|---|---|---|---|---|
| Four devices | R/C 1 | R/C 2 | R/C 3 | EFI |
| Fast | 0.24 | 0.18 | 0.12 | 0.05 |
| Delay | 0.26 | 0.20 | 0.14 | |
| Three devices | R/C 1 | R/C 2 | EFI | |
| Fast | 0.18 | 0.12 | 0.05 | |
| Delay | 0.20 | 0.14 | | |
| Two devices | R/C 1 | EFI | | |
| Fast | 0.12 | 0.05 | | |
| Delay | 0.14 | | | |

## 3. Optimal Protection Coordination and Standardization Methods

*3.1. Objective Function of TDS*

The standardized TCC is expressed by IEC as follows:

$$t = \left[ \frac{A}{\left(\frac{I_f}{I_p}\right)^B - 1} \right] \times TDS \tag{4}$$

where $I_f$ is the fault current and $I_p$ is the pick-up current (minimum operating current). $A$ and $B$ are coefficients according to the type of TCC. The existing TCC equation has a non-linear relationship between $t$ and the fault current/pick-up current ($I_f/I_p$) [20]. Table 4 describes the parameters of the time–current characteristic curves for overcurrent relays [21].

**Table 4.** Characteristics of the time–current characteristic curve.

| Parameter | Curve Type | | | | |
|:---:|:---:|:---:|:---:|:---:|:---:|
| | Short-Time Inverse | Standard Inverse | Very Inverse | Extremely Inverse | Long-Time Inverse |
| A | 0.04 | 0.14 | 13.5 | 80 | 120 |
| B | 0.04 | 0.02 | 1 | 2 | 1 |

Based on the condition analysis in Section 2, the TDS for each protective device is optimized in consideration of linearization, constraints and the objective function of the IEC-VI protection coordination characteristic curve equation. In an actual distribution system, when a protective device is installed, the distance between the protective device and the substation is fixed, which leads to constant line impedance. Therefore, the fault currents of a short or ground fault (same earth impedance) are identical. In addition, the pick-up current is set by the protection coordination operator according to certain guidelines. Therefore, if a protective device is specified, the fault current and the pick-up current can be considered constants. Consequently, the TCC curve is transformed into a linear function, and the nonlinear optimization problem can be redefined linearly to $\alpha TDS$, where $\alpha$ is a constant.

The linear optimization method aims to obtain a TDS that minimizes the sum of blocking operating times in each protective device. The objective function to obtain the appropriate TDS value of a distribution line in which four protective devices are connected in series was defined as follows:

$$\min_{T_{11}, T_{22}, T_{33}, T_{44}} \{T_{11} + T_{22} + T_{33} + T_{44}\} \tag{5}$$

where $T_{nk}$ means the blocking operation time of R/C $k$ when $F_n$ failure occurs. $F_n$ means a single-line ground or three-phase short fault that occurred between the $n$ and $n + 1$-th protective devices. For example, the fast blocking operation time of R/C 1 when $F_1$ failure occurs ($T_{11}$) for a three-phase short circuit fault is calculated as follows:

$$T_{11} = \frac{13.5}{\left(\frac{I_{F1}}{I_{pR/C1}}\right) - 1} \times 60 \times TDS_1 \qquad [\text{cycles}] \tag{6}$$

*3.2. Linear Optimization of TDS with Overhead Line Distribution Model Simulation*

Distribution lines in South Korea are constructed and operated within 10 km of each other. In addition, utility poles are required to be spaced 30 to 50 m apart. The shorter the length of a distribution line of the same wire type and dimension, the smaller the line impedance, resulting in a larger fault current. The distribution line model is simulated to have a smaller $I_f/I_p$ ratio with the maximum length of distribution lines operated in South Korea (10 km). The number of protective devices that can be operated in series as much as possible (3 R/C + 1 EFI) was applied to assume the most difficult condition to secure the CTI between protective devices. Therefore, the distribution line model was set up in PSCAD/EMTDC to ensure that the spacing between the four protective devices is 2.5 km based on a 10 km distribution line. If the spacing of the protective devices is uneven and within 2.5 km, or if the spacing is extremely close, the magnitude of the fault current directly under the two protective devices becomes almost the same. In this case, there is a possibility of a simultaneous trip, so the minimum operating current difference between protection devices is limited to 10~20%. The fault currents for each three-phase short and single-line ground ($F_1$, $F_2$, $F_3$, $F_4$) are calculated at each protective device. Figure 6 shows the standardized cable model for the distribution line. The distribution line model uses 22.9 kV, ACSR 160 mm$^2$, with a length of 10 km. The total length is 10 km, and 2.5 km

is equally divided between the four protective devices. The equivalent impedance of the simulation cable model is described in Table 5.

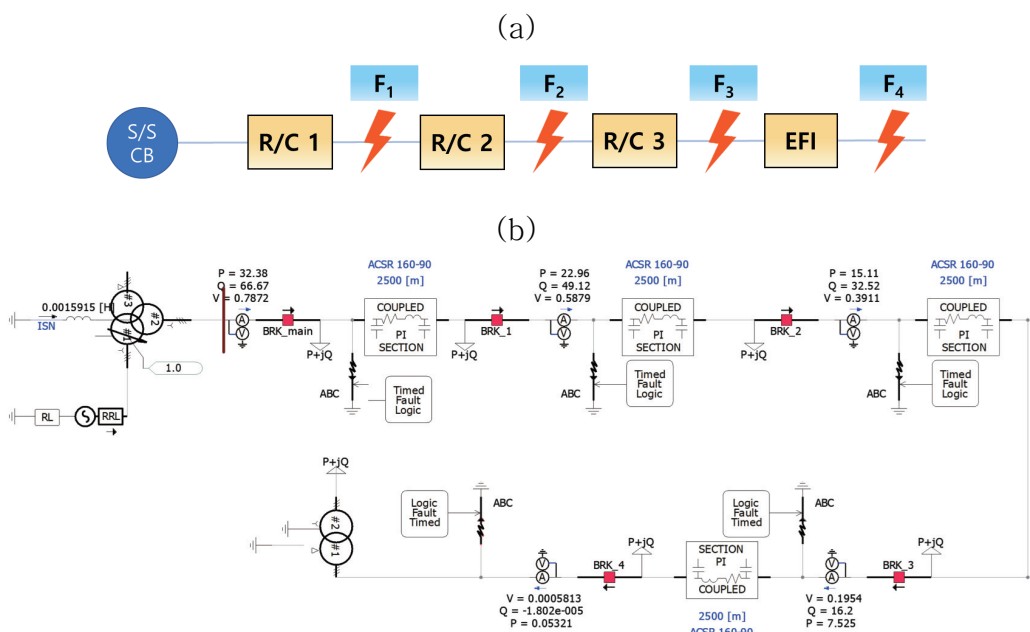

**Figure 6.** Short circuit, ground fault current review model: (**a**) Diagram of power distribution line and (**b**) the fault simulation model based on PSCAD.

**Table 5.** Equivalent impedance of distribution line model.

|  | $Z_1$ [p.u.] | $Z_2$ [p.u.] | $Z_0$ [p.u.] |
|---|---|---|---|
| Source Impedance (@ 100 MVA) | 0.00105 + j0.01146 | | 0.00527 + j0.029 |
| CNCV/W Impedance (@ 100 MVA) | 0.01823 + j0.028222 | | 0.053203 + j0.016495 |
| ACSR Impedance (@ 100 MVA) | 0.034999 + j0.077498 | 0.086824 + j0.228644 | 0.091004 + j0.22841 |
|  | $X_{HM}$ [p.u.] | $X_{ML}$ [p.u.] | $X_{LH}$ [p.u.] |
| MTr Impedance (@ 60 MVA) | j0.14496 | j0.0669 | j0.2538 |

The fault current simulation results for each protective device are described in Table 6. Therefore, the blocking operating time for the TDS when a defect occurs at each point ($T_nk$) can be calculated.

**Table 6.** Fault current simulation results for each protective device.

| Fault Type | CB Draw-Out | $I_{F1}$ | $I_{F2}$ | $I_{F3}$ | $I_{F4}$ |
|---|---|---|---|---|---|
| Three-phase short circuit [A] | 7493 | 5683 | 3821 | 2880 | 2305 |
| Single-line ground circuit [A] | 4680 | 2278 | 1497 | 1110 | 878 |

The pick-up current for each fault in the simulation model was set as Table 7. The minimum operating current is selected by considering the load side conditions of each protective device. Therefore, based on Tables 6 and 7, the fault current and the pick-up current can be calculated for faults at each location.

In order to obtain an appropriate TDS value, the conditions are summarized in Table 8 below to define the constraints for linear optimization.

**Table 7.** Pick-up current values of protective device.

| Type | | CB Ry | R/C 1 | R/C 2 | R/C 3 | EFI |
|---|---|---|---|---|---|---|
| Three-phase Short | Fast [A] | 540 | 430 | 290 | 220 | 184 |
| | Delay [A] | 480 | 440 | 290 | 230 | 184 |
| Single-line Ground | Fast [A] | 230 | 180 | 110 | 85 | 40 |
| | Delay [A] | 220 | 175 | 114 | 88 | 40 |

**Table 8.** Constraints for each TDS selection.

| | Type | Condition |
|---|---|---|
| 1 | Blocking operation time | IEC-VI curve: $t = \dfrac{13.5}{\left(\frac{I_f}{I_p}\right)^1 - 1} \times 60 \times TDS$  [cycles] |
| 2 | Measurement error 5P20 | Rated (600 A) $\pm$1%, up to 20 times (12,000 A) $\pm$10% Within 6500 A of short-circuit fault/Within 5800 A of ground fault |
| 3 | $I_p$ difference between protective devices | Within 10~20% |
| 4 | Substation TDS range | Short: above 3.78 Ground: above 4.0 |

Consequently, the objective function of each fault case with CTI constraints based on Equation (5) and Table 1 are summarized in Table 9.

**Table 9.** Conditions for defining constraints.

| | | | |
|---|---|---|---|
| Objective function | Three-phase short circuit | Fast | $66.3050TDS_1 + 66.5251TDS_2 + 66.9925TDS_3 + 3.5154TDS_4$ |
| | | Delay | $67.9763TDS_1 + 66.5250TDS_2 + 70.3019TDS_3 + 3.5134TDS_4$ |
| | Single-line ground fault | Fast | $69.4948TDS_1 + 66.5250TDS_2 + 70.3019TDS_3 + 1.9332TDS_4$ |
| | | Delay | $67.4037TDS_1 + 66.7679TDS_2 + 69.7456TDS_3 + 1.9332TDS_4$ |
| Minimum CTI constraints | | | CB Ry-R/C: 167ms (10 cycles) R/C-R/C: 58ms (3.5 cycles) R/C-EFI: 58ms (3.5 cycles) |
| Constraints | Three-phase short circuit | Fast | $33.8587TDS_S - 66.3050TDS_1 \geq 10$ cycles $102.7131TDS_1 - 66.5251TDS_2 \geq 3.5$ cycles $90.6950TDS_2 - 66.9925TDS_3 \geq 3.5$ cycles $85.4676TDS_3 - 3.5134TDS_4 \geq 3.5$ cycles |
| | | Delay | $31.9402TDS_S - 67.9763TDS_1 \geq 10$ cycles $105.4126TDS_1 - 66.5250TDS_2 \geq 3.5$ cycles $90.6950TDS_2 - 70.3019TDS_3 \geq 3.5$ cycles $89.7831TDS_3 - 3.5134TDS_4 \geq 3.5$ cycles |
| | Single-line ground fault | Fast | $37.0764TDS_S - 69.4948TDS_1 \geq 10$ cycles $110.7062TDS_1 - 64.2394TDS_2 \geq 3.5$ cycles $89.1000TDS_2 - 67.1707TDS_3 \geq 3.5$ cycles $86.8222TDS_3 - 1.9332TDS_4 \geq 3.5$ cycles |
| | | Delay | $36.1484TDS_S - 67.4037TDS_1 \geq 10$ cycles $107.2239TDS_1 - 66.7679TDS_2 \geq 3.5$ cycles $92.7108TDS_2 - 69.7456TDS_3 \geq 3.5$ cycles $90.2278TDS_3 - 1.9332TDS_4 \geq 3.5$ cycles |
| Common constraint | | | $0.10 \leq TDS \leq 2.00$, 0.01 step |

Based on the constraints and objective functions, linear optimization based on dual-simplex is performed, and the optimal TDS value of each protective device is calculated for each failure situation and operation (Fast, Delay), which is described in Table 10.

The optimal value is the minimum cycle summation of the operation time in case of failure ($F_1$, $F_2$, $F_3$, $F_4$) under each protective device. Through the above process, the optimal TDS value of each protective device can be derived.

**Table 10.** Linear optimized TDS value.

| | | $TDS_1$ (R/C 1) | $TDS_2$ (R/C 2) | $TDS_3$ (R/C 3) | $TDS_4$ (EFI) | Optimal Value (Cycles) |
|---|---|---|---|---|---|---|
| Three-phase Short circuit | Fast | 0.2172 | 0.1623 | 0.1075 | | 32.4 |
| | Delay | 0.2295 | 0.1804 | 0.1195 | 0.05 | 36.0 |
| Single-line ground fault | Fast | 0.2072 | 0.1681 | 0.1072 | | 32.4 |
| | Delay | 0.2314 | 0.1797 | 0.1204 | | 36.0 |

*3.3. Protection Cooperation Method Using Proposed TDS and MRT*

The linearly optimized TDS and MRT in which two to four protective devices are configured in series are presented in Table 11. The optimized TDS and MRT can be applied in accordance with the determined order by intuitively judging the serial configuration of the protective device. Since reclosing is prohibited in underground distribution lines due to the risk of human safety hazards, only the Fast TDS and MRT were applied, not the sequence operation of 2F2D.

**Table 11.** Optimized TDS and MRT for each protective device.

| Four Devices | | R/C 1 | R/C 2 | R/C 3 | EFI |
|---|---|---|---|---|---|
| Fast | TDS | 0.21 | 0.16 | 0.10 | |
| | MRT | 0.24 | 0.18 | 0.12 | 0.05 |
| Delay | TDS | 0.23 | 0.18 | 0.12 | |
| | MRT | 0.26 | 0.20 | 0.14 | |
| Three Devices | | R/C 1 | R/C2 | EFI | |
| Fast | TDS | 0.16 | 0.10 | | |
| | MRT | 0.18 | 0.12 | 0.05 | |
| Delay | TDS | 0.18 | 0.12 | | |
| | MRT | 0.20 | 0.14 | | |
| Two Devices | | R/C 1 | EFI | | |
| Fast | TDS | 0.10 | | | |
| | MRT | 0.12 | 0.05 | | |
| Delay | TDS | 0.12 | | | |
| | MRT | 0.14 | | | |

TCCs based on IEC-VI applying the proposed TDS and MRT in the distribution line with four protective devices is shown in Figure 7. The TCCs satisfied all the operating characteristics of CTI and 2F2D for each protective device. Table 12 depicts the blocking time results for each fault of the overhead power distribution line including four series-connected protective devices.

In the case of an underground line failure, a large-scale fault current is instantaneously generated, so only one substation CB and MCA can partake in coordination protection. Therefore, the same settings as EFI were applied to MCA in underground distribution lines. In the overhead–underground mixed line, unlike the underground line, protection coordination is required between the MCA and load-side R/Cs. The results of the protection coordination operation time of protective devices are shown in Table 13. A 5P20

measurement error of 10% and a pick-up current difference of 20% were applied. Results verify the feasibility of accurate protection coordination between protective devices in all failure situations.

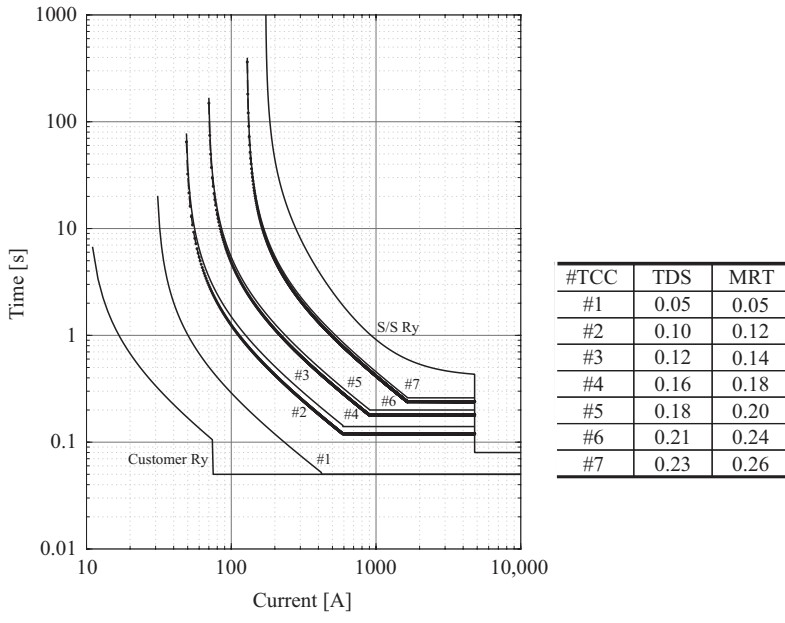

**Figure 7.** TCC curves based on proposed algorithm.

**Table 12.** Result of protection coordination using proposed algorithm for the case of three-phase short circuit and single-line ground fault.

| | | | | | | |
|---|---|---|---|---|---|---|
| **Three-phase short circuit** | Pick-up current [A] | 384 | 345 | 276 | 220 | 184 |
| | CTI Fast [ms] | | | 60 | 60 | 59 |
| | CTI Delay [ms] | | 179 | 60 | 60 | |
| **Single-line ground circuit** | Pick-up current [A] | 84 | 75 | 60 | 48 | 40 |
| | CTI Fast [ms] | | | 60 | 60 | 66 |
| | CTI Delay [ms] | | 176 | 60 | 60 | |

**Table 13.** Result of protection coordination using proposed algorithm in overhead-underground mixed line.

| Pick-up current [A] | 384 | 345 | 276 | 220 | 184 |
|---|---|---|---|---|---|
| CTI Fast [ms] | | | 60 | 60 | 59 |
| CTI Delay [ms] | | 179 | 60 | 60 | |

## 4. Analysis of Actual Distribution Line Application Results

The proposed algorithm was applied to the actual power distribution system to verify the performance of protection cooperation. The distribution line consisted of three protective devices in series (R/C 1, R/C 2, and EFI). Figure 8 illustrates the diagram of the actual distribution system with a total length of 9.27 km for fault simulation and protection coordination algorithm verification. The fast and delay TDS and MRT settings for N.S 2 R/C 1, N.S 10 R/C 2, and J.S 11 EFI were used in the three protective devices case in Table 11. The pick-up currents of the protective devices are 40, 25 and 15 A respectively. The single-line ground fault (100 Ω) was simulated by an artificial fault generator (AFG) which was installed at the end of the line.

After simulating the fault using AFG in real systems, the fault voltage and current of each circuit breaker and whether or not the circuit breaker operates normally are summa-

rized in Table 14 and Figure 9. Consequently, R/C 2 operated normally and blocked the fault section, while R/C 1 did not operate. The results show that protection cooperation is successfully operated even with uneven distribution of protective devices using the proposed algorithm. Based on the results of this study, the proposed method has been recently implemented for approximately 12,143 overhead and underground protective devices in South Korea.

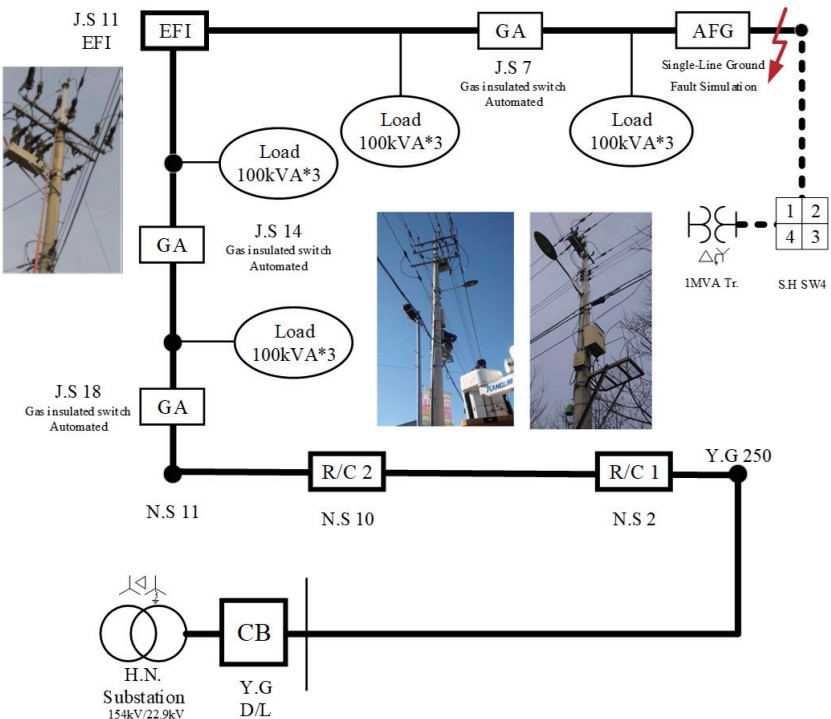

**Figure 8.** Actual power distribution system diagram.

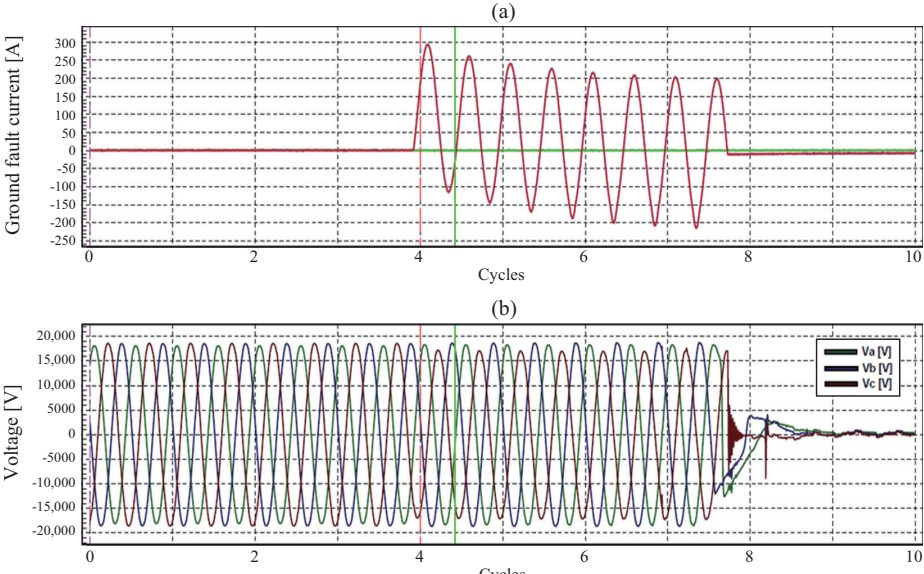

**Figure 9.** Single-line ground fault simulation and interception test results: (**a**) simulated fault current and (**b**) phase voltage results.

**Table 14.** Result of protection coordination using proposed algorithm in overhead–underground mixed line.

| Fault Voltage [V] | | Fault Current [A] | | Breaker Operation | |
| --- | --- | --- | --- | --- | --- |
| | | | | R/C 2 | EFI |
| A | 13,277 | A | 0 | | |
| B | 13,440 | B | 0 | | |
| | | | | X | O |
| C | 12,981 | C | 146 | | |
| $3I_0$ | 119 | $3V_0$ | 146 | | |

## 5. Conclusions

In this study, a standardized TCC design method was presented through the serial connection of protective devices located on distribution lines. The proposed algorithm calculates the optimal TCC for each protective device by considering the analysis of the appropriate minimum operating current, fault current and CTI constraints in a series-connected distribution system. The accuracy of protection cooperation was improved by defining the characteristics of the protective devices and CTI. In addition, by applying the measurement error standard, protection coordination may be applied even at an error rate of 20 times the rated current. The pick-up current difference for distribution line protection coordination was calculated and the appropriate TDS was derived through linear optimization. Through the application of MRT, the protection coordination is verified to apply in all possible failure cases. Therefore, standardized TDS/MRT were presented in order to set the same conditions for all distribution lines.

The proposed method is expected to significantly reduce the burden of the distribution protection coordinators as TDS and MRT can be applied immediately after checking the serial configuration of the distribution line protective devices. The proposed algorithm is able to contribute substantially to the active protection cooperation and automation design of the next-generation advanced distribution management system (ADMS) to be applied in the future.

**Author Contributions:** Conceptualization, G.K. and C.-K.L.; methodology, G.K. and W.-H.K.; software, W.-H.K., formal analysis, G.K.; data curation, W.-H.K. and G.K.; writing—original draft preparation, G.K.; visualization, G.K.; supervision, C.-K.L. and G.K. All authors have read and agreed to the published version of the manuscript.

**Funding:** This work has supported by the National Research Foundation of Korea (NRF) grant funded by the Korea government (MSIT) (No. NRF-2021R1G1A1093407).

**Conflicts of Interest:** The authors declare no conflict of interest. The funding sponsors had no role in the design of the study; in the collection, analyses, or interpretation of data; in the writing of the manuscript, and in the decision to publish the results.

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
