# Peer review of "Selection of TCC Curve and Protection Cooperation Method of Distribution Line Using Linear Optimization"

_electronics, doi:10.3390/electronics12122705_

Round 1

Reviewer 1 Report

1) Table 4 and Figures 5 and 7 should be placed after their first mention in the text.
2) Could you please explain in detail the target functions in Table 7? How were the coefficients derived?
3) In Section 4, you are talking about the proposed algorithm application for various cases and even about the actual distribution line. But there are no description, data, scheme or line parameters in the text, only results of the proposed optimization algorithm. Add the electrical scheme with some text explanation for considered cases in the results presentation part is suggested.

Reviewer 2 Report

In this work, an interesting adaptive time-current curve method by selecting the time dial setting and minimum response time of individual protective devices using linear optimization technique has been proposed. 

The abstract is accurate but some results and obtained data must be included. 

In lines 191-195, the three-phase fault currents were stated in the range 7,000∼8,000 A and you have chosen the value of 6.5kA. It is not more accurate to simulate the worst case when the current is the highest? In the figure 2, you have presented values near 10kA and therefore, your estimation is out of range. Please consider how to select that values to include all the systems under study. The same for one-line ground fault current.

In lines 241-246 you have selected 60ms for all and you have included an additional in the end-line. Why 60? Can you provide more reasons and justify the selection?

Can you provide an enlarged figure 5 (bottom)? It is difficult to see the written data.

In each figure and table caption, please note that the sentence must be ended with a point. 

Even when PSCAD is a well-known software, a subsection explaining the software, where you have stated the methods and constrains, are useful to the readers. I recommend to include a brief paragraph about it. 

Section 4 should be improved because some results have been not explained. In the conclusions, please, compare your results with your method with the literature, advantages, disadvantages, implications, among others. 

Round 2

Reviewer 2 Report

Thank you for addressing the proposed changes.